# The effects of *Cstb* duplication on APP/amyloid-β pathology and cathepsin B activity in a mouse model

Yixing Wu[1], Heather T. Whittaker[2], Suzanna Noy[2], Karen Cleverley[2], Veronique Brault[3], Yann Herault[3], Elizabeth M. C. Fisher[2,4], Frances K. Wiseman[1,4]*

1 UK Dementia Research Institute at UCL, London, United Kingdom, 2 Department of Neuromuscular Diseases, UCL Institute of Neurology, London, United Kingdom, 3 Université de Strasbourg, CNRS, INSERM, Institut de Génétique Biologie Moléculaire et Cellulaire (IGBMC), Illkirch, France, 4 LonDownS Consortium, London, United Kingdom

* f.wiseman@ucl.ac.uk

**Data Availability Statement:** The original western blotting images are available in Figshare: https://doi.org/10.6084/m9.figshare.14434895.v1. The

## Abstract

People with Down syndrome (DS), caused by trisomy of chromosome 21 have a greatly increased risk of developing Alzheimer's disease (AD). This is in part because of triplication of a chromosome 21 gene, *APP*. This gene encodes amyloid precursor protein, which is cleaved to form amyloid-β that accumulates in the brains of people who have AD. Recent experimental results demonstrate that a gene or genes on chromosome 21, other than *APP*, when triplicated significantly accelerate amyloid-β pathology in a transgenic mouse model of amyloid-β deposition. Multiple lines of evidence indicate that cysteine cathepsin activity influences APP cleavage and amyloid-β accumulation. Located on human chromosome 21 (Hsa21) is an endogenous inhibitor of cathepsin proteases, *CYSTATIN B* (*CSTB*) which is proposed to regulate cysteine cathepsin activity *in vivo*. Here we determined if three copies of the mouse gene *Cstb* is sufficient to modulate amyloid-β accumulation and cathepsin activity in a transgenic *APP* mouse model. Duplication of *Cstb* resulted in an increase in transcriptional and translational levels of *Cstb* in the mouse cortex but had no effect on the deposition of insoluble amyloid-β plaques or the levels of soluble or insoluble amyloid-β42, amyloid-$\beta_{40}$, or amyloid-$\beta_{38}$ in 6-month old mice. In addition, the increased CSTB did not alter the activity of cathepsin B enzyme in the cortex of 3-month or 6-month old mice. These results indicate that the single-gene duplication of *Cstb* is insufficient to elicit a disease-modifying phenotype in the dupCstb x tgAPP mice, underscoring the complexity of the genetic basis of AD-DS and the importance of multiple gene interactions in disease.

## Introduction

Alzheimer's Disease (AD) is the most common neurodegenerative disorder [1]. Accumulation of amyloid-β (Aβ) plaques and formation of hyperphosphorylated tau neurofibrillary tangles are pathological hallmarks of AD [2]. People with Down syndrome (DS), a genetic disorder caused by chromosome 21 (Hsa21) trisomy develop the characteristic features of AD

minimal data set underlying the study is available in the paper.

**Funding:** Y. W. is funded by an Alzheimer's Research UK Senior Research Fellowship held by F.K.W (ARUK-SRF2018A-001). https://www.alzheimersresearchuk.org/research F.K.W. is also supported by the UK Dementia Research Institute (UKDRI-1014) which receives its funding from DRI Ltd, funded by the UK Medical Research Council, Alzheimer's Society and Alzheimer's Research UK. https://ukdri.ac.uk/ https://mrc.ukri.org/ https://www.alzheimersresearchuk.org/research https://www.alzheimers.org.uk/ F.K.W. and E.M.C.F. also received funding that contributed to the work in this paper from the MRC via CoEN award MR/S005145/1. https://www.coen.org/ https://mrc.ukri.org/ E.M.C.F. received funding from a Wellcome Trust Strategic Award (grant number: 098330/Z/12/Z) awarded to The London Down Syndrome (LonDownS) Consortium (E.M.C.F) and a Wellcome Trust Joint Senior Investigators Award (grant numbers: 098328, 098327). https://wellcome.org/ The funders had no role in study design, data collection and analysis, decision to publish, or preparation of the manuscript.

**Competing interests:** F.K.W. has undertaken consultancy for Elkington and Fife Patent Lawyers unrelated to the work in the manuscript and is also a PLoS One Academic Editor. This does not alter our adherence to PLoS ONE policies on sharing data and materials.

pathology by the age of 50 [3, 4] and by the age of 60 approximately 2/3 of individuals will have developed the clinical features of dementia [5]. The amyloid precursor protein (APP) is encoded by a Hsa21 gene, *APP*, and is cleaved and processed to form amyloid-β (Aβ) which then aggregates to form plaques [6, 7]. Duplication of *APP* is sufficient to cause early-onset AD [8, 9] and in the absence of three copies of *APP* people with DS do not develop early onset AD [10, 11]. However, whether duplication of other Hsa21-located genes also contributes to the pathogenesis of AD in DS remains unknown.

Recently, by crossing a trisomic Hsa21 mouse model (Tc1), which has an extra copy of 75% Hsa21 genes but lacks an additional functional copy of *APP*, with an APP-amyloid deposition mouse model (J20-tgAPP) [12–14], we found that three copies of Hsa21 genes other than *APP* exacerbate Aβ deposition and cognitive deficits [15], indicating that other Hsa21 genes may also play important roles in the pathogenesis of AD-DS. In order to investigate the potential contribution of other Hsa21 genes to DS phenotypes, partial trisomy DS mouse models, which contain a segmental duplication of mouse chromosome regions or genes that are syntenic to Hsa21 have been generated (S1 Fig) [16].

A potential candidate gene on Hsa21 is *CSTB*, which encodes cystatin B (CSTB), an endogenous inhibitor of cystine proteases [17]. Human CSTB has also been proposed to be an interacting partner of Aβ and colocalises with intracellular inclusions of Aβ in cultured cells [18]. Protein levels of CSTB have been reported to be increased in the brain of people who have AD [19]. Knocking out *Cstb* by crossing the *Cstb*[tm1Rm] [20] and an *APP* transgenic mouse model (TgCRND8) increased cathepsin activity [21] and reduced Aβ aggregation [21], indicating that this gene could have a role in Aβ pathogenesis. Several studies suggest that cathepsin B influences the development of Aβ pathology. However, the dominant mode of action is unclear. In secretory vesicles of neuronal chromaffin cells, cathepsin B inhibition disrupted the conversion of endogenous APP to Aβ [22]. Treating an *APP* transgenic model (Tg(THY1-APP)2Somm) that expresses wildtype APP with a cathepsin B inhibitor CA074 Me leads to reduced Aβ levels and improved memory [23, 24]. In contrast knocking down cathepsin B in mice expressing familial AD-mutant human APP leads to increased levels of $A\beta_{42}$ and plaque deposition, indicating an anti-amyloidogenic role of cathepsin B [25]. Consistent with this, increasing Cathepsin B activity by treating with PADK or AAV-mediated overexpression leads to reduction in $A\beta_{42}$ and plaque formation [26–28]. Thus, the effect of *Cstb* on Aβ accumulation may be mediated by a direct effect of the endogenous inhibitor on cathepsin B activity [21]. The increase in gene copies of *Cstb* that occurs in people who have DS, may cause a decrease in cathepsin activity in the brain and elevation of Aβ levels and plaque accumulation or a modulation of APP processing. Whether the three-copies of *CSTB* affects AD-DS pathogenesis is unknown.

To determine if 3-copies of *CSTB* could influence the pathogenesis of AD in people with DS we crossed the J20 tgAPP mouse model of amyloid-β deposition to a mouse with a heterozygous duplication of the *Cstb* locus on Mmu10 [29]. We found that duplication of *Cstb* increased transcriptional and translational levels of CSTB in the brain. However, this did not lead to changes in APP processing, plaque deposition or Aβ levels. Duplication of *Cstb* did not change the activity of cathepsin B in the cortex at 3-months or 6-months of age.

## Results

### *Cstb* mRNA and protein levels increased by duplication of *Cstb*

To verify if elevated expression of *Cstb* occurs in mice with a duplicated *Cstb* locus, the amount of *Cstb* mRNA in the cortex was determined. Duplication of *Cstb* significantly increased the transcriptional level of *Cstb* (Fig 1A). Presence of the tgAPP had no effect on *Cstb* mRNA

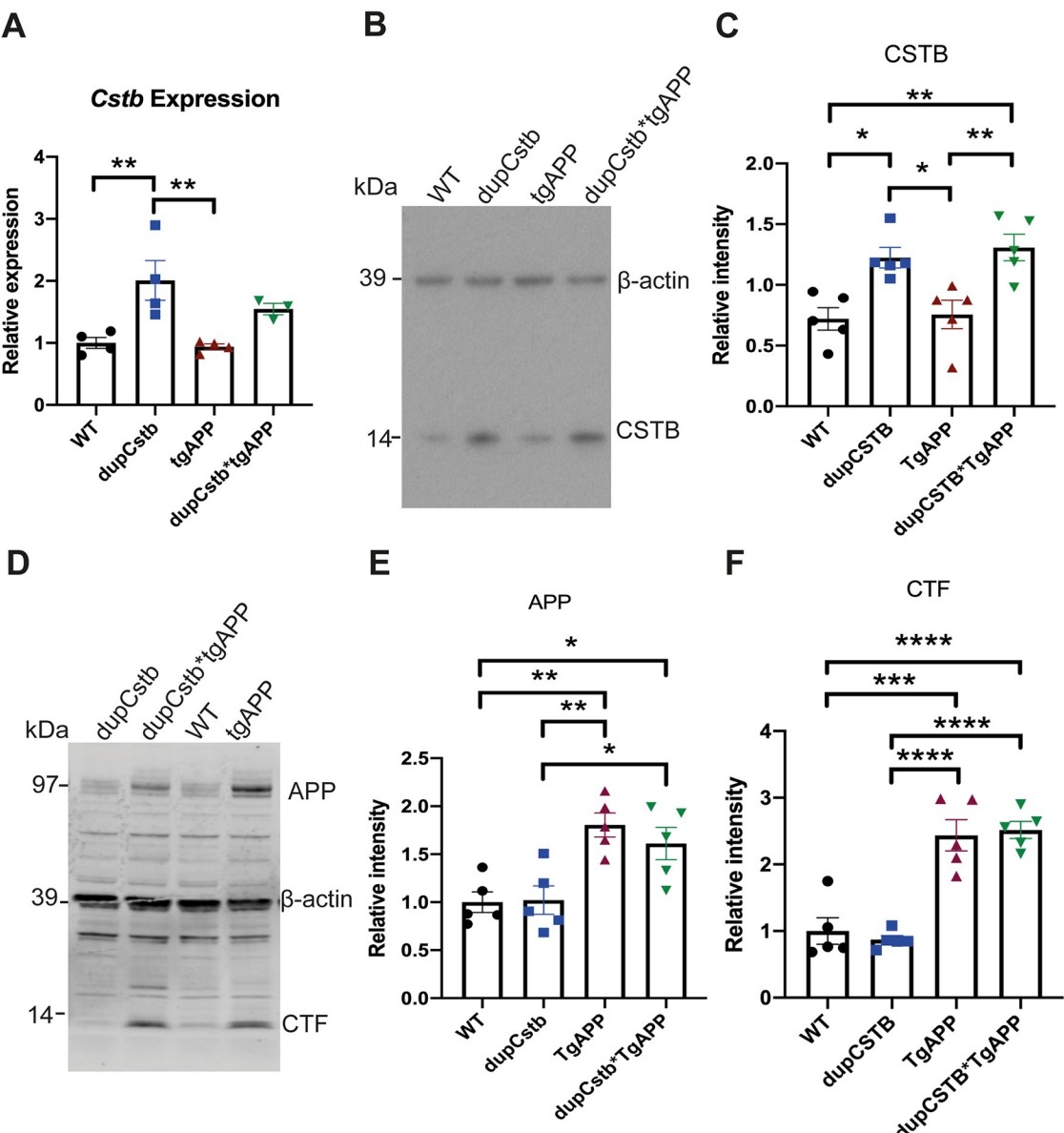

**Fig 1. Transcriptional and translational levels of cortical CSTB in 3-month-old mice.** (A) Levels of *Cstb* mRNA in the cortex of 3-month old mice. Data are relative to *Actb* and *Gapdh* housekeeping genes, and represented as mean expression levels for mice of each genotype ± SEM. Duplication of *Cstb* caused an increase in the relative levels of *Cstb* mRNA (F(1,7) = 7.240, p = 0.031). No effect of sex, tgAPP, or interaction between dupCstb and tgAPP was evident (n = 3–5 per genotype, 9 females and 9 males in total). (B) Representative western blot probed with anti-CSTB and anti-β-actin antibodies. (C) Protein band density was quantified using ImageJ, normalised to β-actin, and is shown as fold-change relative to wildtype levels (mean ± SEM), n = 5 for each genotype (9 females and 11 males in total). (D) Representative western blot probed with anti-APP C-Terminal Fragment (CTF) and anti-β-actin antibodies. (E) APP protein band density was quantified using ImageJ, normalised to β-actin, and is shown as fold-change relative to wildtype levels (mean ± SEM), n = 5 for each genotype (9 females and 11 males in total). (F) CTF protein band density was quantified using ImageJ, normalised to β-actin, and is shown as fold-change relative to wildtype levels (mean ± SEM), n = 5 for each genotype (9 females and 11 males in total). Bonferroni post-hoc pair-wise comparisons p < 0.05 = *, p < 0.01 = **, p < .001 = ***, p < 0.0001 = ****.

abundance and no interaction between tgAPP and dupCstb was evident. Sex of the mouse did not affect the level of *Cstb* mRNA (Fig 1A).

To determine whether the increased amount of mRNA caused an increased amount of CSTB protein in dupCstb mice, cortical CSTB protein levels were measured by western

blotting. The presence of a *Cstb* duplication caused a significant increase in CSTB protein (Fig 1B). The levels of CSTB were not changed by presence of the tgAPP or sex of the mouse, and there was no interaction observed between dupCstb and tgAPP. Mice containing dupCstb, including those in the dupCstb cohort and the dupCstb*tgAPP cohort, had approximately 2 times more CSTB protein than mice without dupCstb (Fig 1C). These data show that duplication of *Cstb* increases both the RNA and protein of CSTB in the cortex. Western blot was used to confirm the increase in expression of APP protein in the tgAPP model (Fig 1D). APP and APP C-Terminal Fragment (CTF) levels in the tgAPP group and the dupCstb*tgAPP cohort were significantly increased compared to mice without tgAPP but no difference in abundance between the tgAPP and the dupCstb*tgAPP cohorts was observed (Fig 1E and 1F).

## *Cstb* duplication does not alter plaque deposition at 6 months of age in the cortex or hippocampus

We used the dupCstb*tgAPP mouse model to investigate if increased CSTB influences Aβ deposition in the brain at 6-months of age. The Aβ deposition in the cortex or hippocampus was analysed using a 4G8 monoclonal antibody. Aβ deposition in both brain regions was apparent in mice with the tgAPP and was significantly elevated compared with wildtype controls (Fig 2A and 2B). *Cstb* duplication did not change the level of Aβ deposition, in either the presence of the absence of tgAPP (Fig 2B). The results were also not significantly affected by the sex of the mouse.

## *Cstb* duplication does not alter Aβ levels in the cortex at 6 months of age

We undertook a biochemical assay to determine the quantity and solubility of $A\beta_{38}$, $A\beta_{40}$, and $A\beta_{42}$ in the cortex from 6-month old mice. In both tgAPP and dupCstb*tgAPP groups, there was a differential representation of Aβ isoforms in the soluble and insoluble extracts (Fig 3A–3C). In the Tris fraction $A\beta_{38}$ was below the limit of detection in all but one sample, $A\beta_{38}$ and $A\beta_{40}$ predominating over $A\beta_{42}$ in Triton-extracts (Fig 3B), and the reverse being true for Guanidine hydrochloride extracted fraction (Gnd) (Fig 3C). Additionally, the values of Aβ in the Gnd-extracted fraction were a similar order of magnitude higher than in the Triton-soluble fraction for both genotypes measured. For example, the $A\beta_{42}$ in Gnd extracts was on average 12,154 (± 3,111) pg/mg cortical protein in tgAPP and 9,567 (± 3,133) pg/mg in dupCstb*tgAPP mice, but in Triton-extracts $A\beta_{42}$ was only 14.6 (± 3.19) pg/mg in tgAPP and 10.1 (± 1.45) pg/mg for dupCstb*tgAPP mice. The pattern displayed by both tgAPP and dupCstb*tgAPP groups is consistent with the expected Aβ biochemistry in the J20 mouse brain [30].

To further investigate whether there were any changes in the Aβ peptide abundance due to dupCstb, the ratio of $A\beta_{42}$:$A\beta_{38}$ and $A\beta_{42}$:$A\beta_{40}$ were calculated for all mice carrying the tgAPP. The average ratio is presented for both tgAPP and dupCstb*tgAPP (Fig 3D–3G), and shows that the presence of dupCstb did not have a statistically significant effect on this $A\beta_{42}$:$A\beta_{38}$ or $A\beta_{42}$:$A\beta_{40}$ ratio in the cortex at 6-months of age. Together, these results indicate that *Cstb* duplication does not alter Aβ aggregation or Aβ ratios in the cortex at 6-months of age.

## Cstb duplication does not alter cathepsin B activity in the cortex at 3 or 6 months of age

To explore whether the increase in CSTB protein in dupCstb mouse brain corresponded to functional protease inhibition, a cathepsin B activity assay was conducted on cortex tissue from 3- and 6-month old mice. Activity was measured by the cleavage of a cathepsin B substrate to yield a fluorescence-emitting product. The mean rate of cathepsin B activity in each

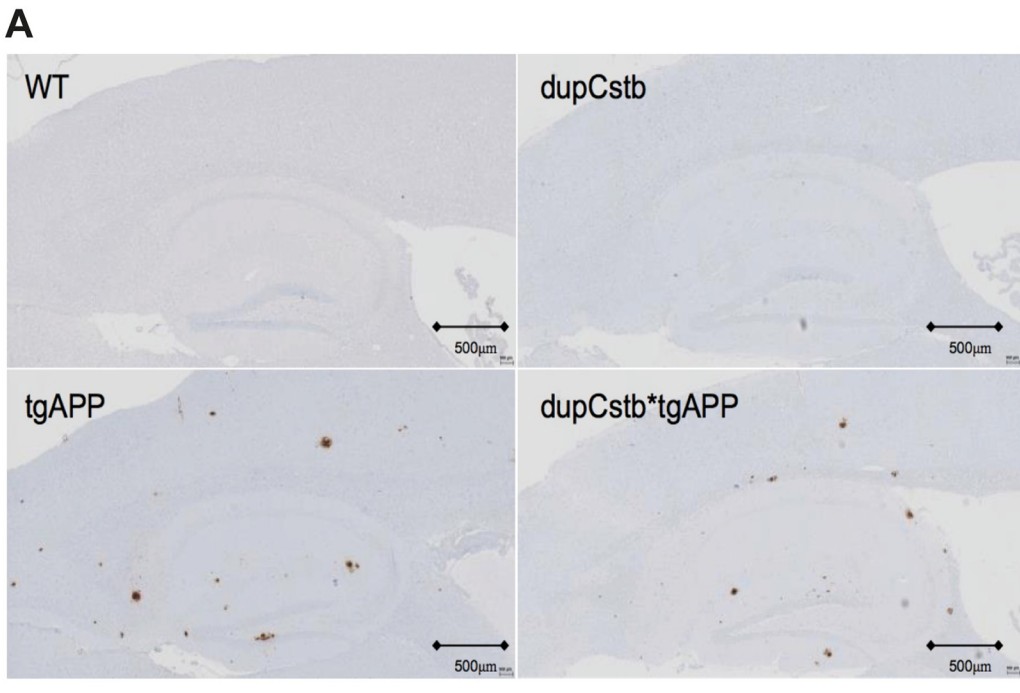

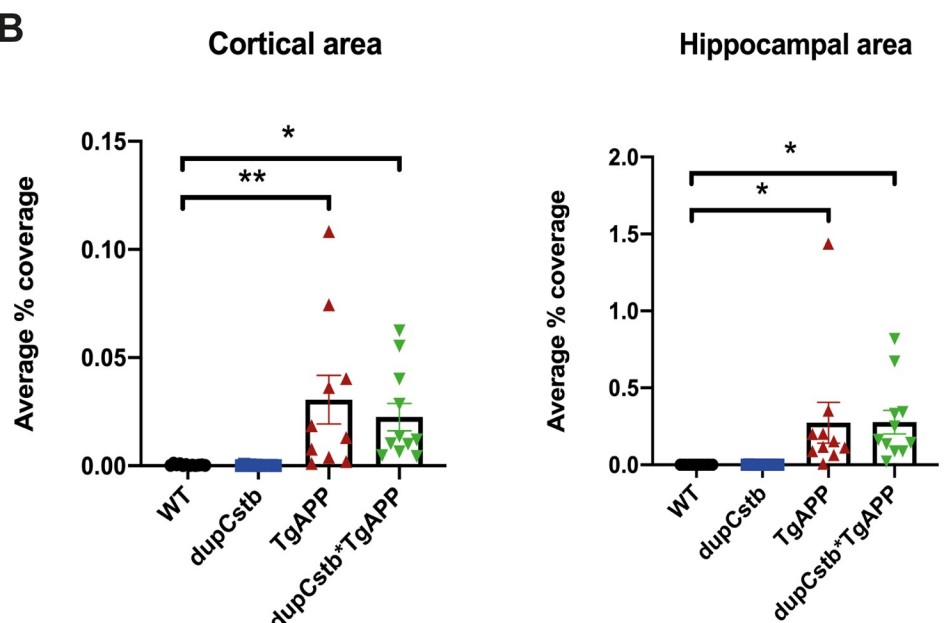

**Fig 2. Aβ plaque deposition in the cortex and hippocampus of 6-month old mice.** (A) Representative images of sagittal sections through the cortex and hippocampus. (B) Area stained with 4G8 antibody (brown) to Aβ was quantified as a percentage of the total area of that region, and group means ± SEM are presented (n = 10–12 per genotype, 23 females and 22 males used in total). Mice with tgAPP had significantly more Aβ staining than those without, in the cortex (F (1,37) = 19.503, p < 0.001) and hippocampus (F(1,37) = 14.466, p = 0.001). There was no significant effect of *Cstb* duplication or sex, and no interaction between tgAPP and dupCstb. Data were analysed using a repeated measures ANOVA. Bonferroni post-hoc pair-wise comparisons p < 0.05 = *, p < 0.01 = **.

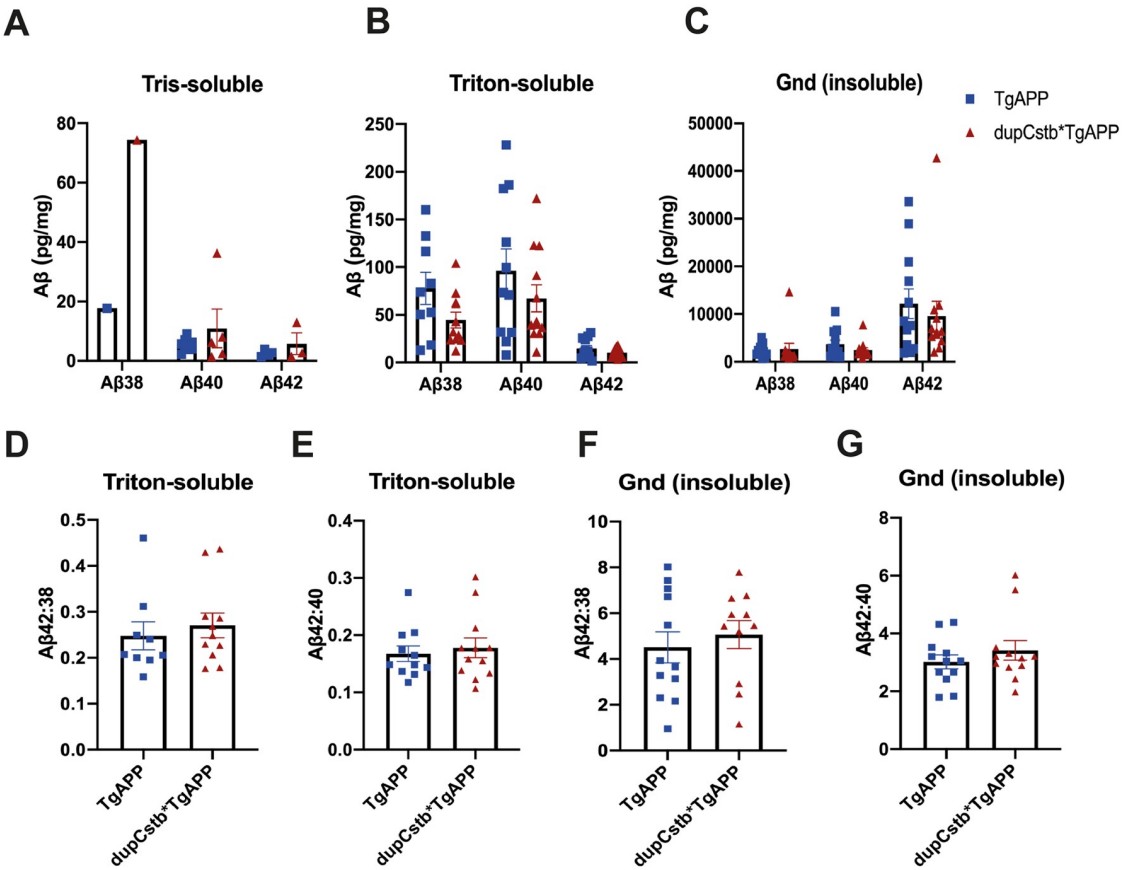

**Fig 3. Cortical Aβ38, Aβ40, and Aβ42 in 6-month old mice, as measured by Meso Scale Discovery Assay.** (A-C) Representation of group means for the Tris-, Triton-, and Gnd-soluble fractions. Samples below detection limit were recorded as '0' in the graph. Values are in pg per mg total protein, ± SEM (WT and dupCstb n = 5 all below limit of detection, tgAPP and dupCstb*tgAPP n = 12; 17 females and 17 males in total). (D-E) Ratio of Aβ42:Aβ38 and Aβ42:Aβ40 was calculated in the Triton- soluble fractions. (F-G) Ratio of Aβ42:Aβ38 and Aβ42:Aβ40 was calculated in the Gnd- soluble fractions. (A-G). No significant differences were found between tgAPP and dupCstb*tgAPP mice by univariate ANOVA.

cohort of littermates as a percentage of the WT mean rate was calculated (Fig 4A and 4B). Univariate ANOVA revealed no significant effect of dupCstb, tgAPP, or sex, and no interaction between these factors, demonstrating that an extra copy of *Cstb* does not affect cathepsin B activity in vivo. To study whether upregulation of *Cstb* affects cathepsin B maturation, western blot analysis was conducted and showed no difference in active cathepsin B/pro-cathepsin B ratio (Fig 4C and 4D), suggesting that the increase in *Cstb* has little impact on cathepsin B maturation in the cortex at 6 months of age.

## Discussion

In this study, we showed that despite a heterozygous duplication of the *Cstb* gene leading to an upregulation of CSTB protein in the mouse brain, this had little impact on APP processing or Aβ plaque deposition in the J20 tgAPP model at 6 months-of-age. The duplication of *Cstb* is not sufficient to alter the APP processing and Aβ plaque formation in this model. In contrast, by crossing the *Cstb* knockout (*Cstb^tm1Rm*) mouse with the tgAPP TgCRND8 model [31], plaque load was significantly reduced at 6-months of age [21], indicating a potential therapeutic importance of *CSTB* knockout. Our data on the insufficient effect of *Cstb* duplication are not

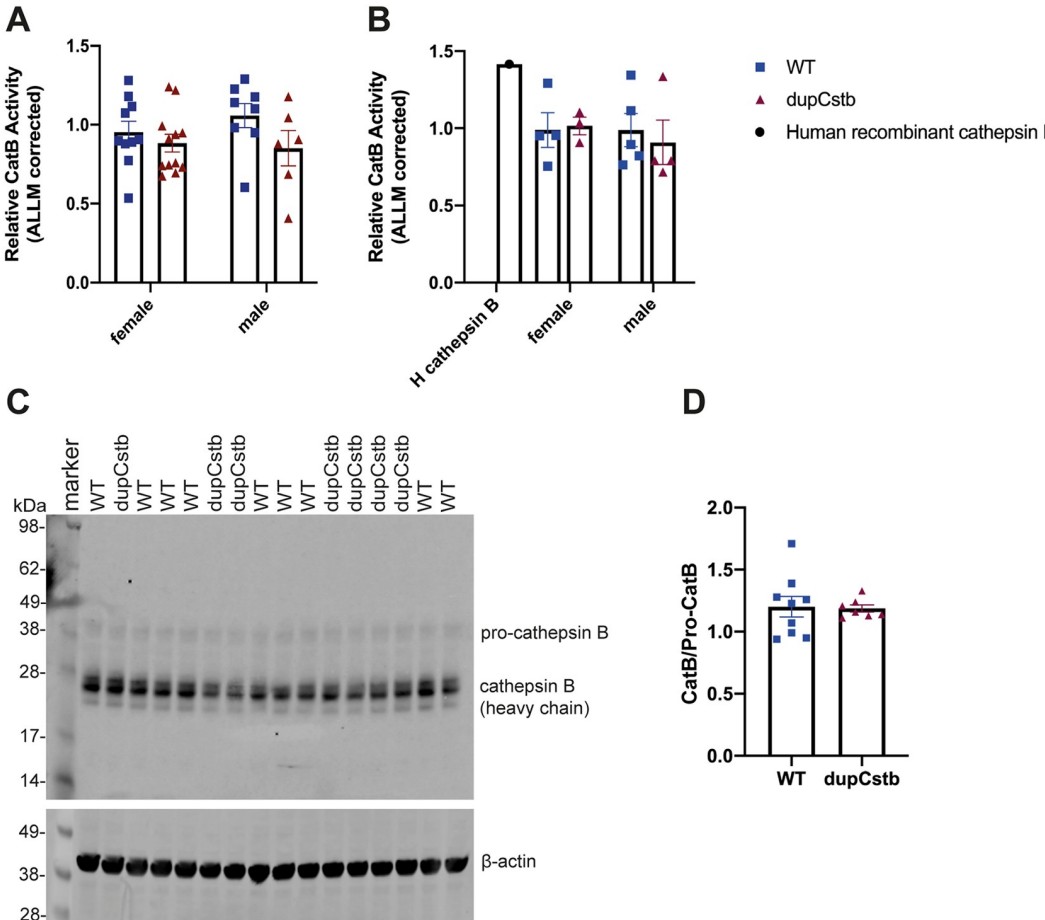

**Fig 4. Activity of cathepsin B in cortical lysates of 3- and 6- month-old mice.** Total cortical proteins were used to determine enzyme activity, as the gain in fluorescence during the linear portion of the reaction curve relative to the wildtype mean. (A) Cortical relative cathepsin B activity at 3-months of age, graphed as group means ± SEM (n = 18 per genotype; (22 females and 14 males in total), revealed no statistically significant effects of dupCstb or sex (univariate ANOVA). (B) Cortical relative cathepsin B activity at 6-months of age, graphed as group means ± SEM, (dupCstb n = 7, wildtype n = 9; 7 females and 9 males in total), revealed no statistically significant effects of dupCstb or sex (univariate ANOVA). Recombinant human cathepsin B (R&D Systems, Cat. No. 953-CY-010) was used as a positive control. (C) Representative western blot probed with an anti-cathepsin B antibody that recognises pro-cathepsin B and cathepsin B heavy chain and an anti-β-actin antibody. (D) Protein band densities of pro-cathepsin B and cathepsin B heavy chain were quantified using ImageJ, normalised to β-actin, and are shown as cathepsin B/pro-cathepsin B ratio in the dupCstb group relative to the ratio in wildtype, graphed as group mean ± SEM (dupCstb n = 7, wildtype n = 9; 7 females and 9 males in total).

necessarily conflicting with the result using the *Cstb^tm1Rm* mouse model, given that loss- of and gain-of-function often have differing outcomes. In order to determine if 3-copies of *CSTB* are necessary to exacerbate Aβ pathology in the context of Hsa21 trisomy, the Tc1*J20 mouse model would need to be crossed with a *Cstb* knockout mouse.

Transgene overexpression in tgAPP mouse models is frequently criticised as being unrepresentative of human familial AD and especially late onset AD [32, 33]. Although some of these mouse models have proven to be valuable reductionist platforms for experimental manipulation, the mutant human gene is expressed at far higher levels than is physiologically relevant to the disease. In this study the high level of APP and robust production of Aβ may not be modifiable by a single additional copy of the *Cstb* gene. Similarly, the results of specific gene 'triplication' studies should be extrapolated with caution to trisomy of Hsa21, in

which transcriptional dysregulation has been documented and may be an effect of aneuploidy in general [34, 35].

Interestingly, localised γ-secretase activity that is selective for lysosomal substrates has been shown to generate an intracellular pool of Aβ42 [36] and this pool may be affected in a model of lysosomal dysregulation. CSTB is localised in the nucleus, cytosol and lysosome [37]. Although it is an endogenous inhibitor of cystine cathepsins [38], the subcellular distribution of CSTB is important in determining and maintaining its regulatory role. Whether the increased level of CSTB in our model is lysosome associated is unknown and if enzymatic activity in the lysosome specifically is reduced has not been determined. However, in this study, the lack of significant changes in soluble Aβ or altered Aβ ratios suggests there are no changes to intracellular Aβ biology in the presence of an extra copy of *Cstb* but further experimentation is required to verify this. We note that early intracellular depositions of amyloid-β is a prominent feature of the earliest stages of AD-DS [39–42].

Cathepsin B enzymatic activity was found to be elevated in brains of *Cstb* knockout mice [21]. However, our study was unable to demonstrate the inverse at 3- or 6-months of age. Aging may modulate the effect of *Cstb* duplication on cathepsin B enzymatic activity, indeed previous studies have suggested that aging exacerbates the effect of trisomy [43]. A further study would be required to investigate this. Of note, under physiological conditions, regulatory sites of cathepsin B may already have been fully saturated by endogenous CSTB, this may underlie the dosage insensitivity of cathepsin B activity to raised *Cstb* gene dose.

A previous study has demonstrated that Cathepsin D activity is decreased in a preclinical model of DS and that Cathepsin B activity was slightly increased. In this study changed Cathepsin D activity was linked with raised APP-CTF [44]. These data are consistent with the results presented here; that 3-copies of *Cstb* (as occurs in the DS patient fibroblasts preclinical model used in Jiang et al. 2019) are not sufficient to markedly change cathepsin B activity. Moreover, here we demonstrate that 3-copies of *Cstb* are not sufficient to alter APP or APP-CTF abundance *in vivo*, indicating that APP processing is not altered. Overall, our study reveals that duplication of the *Cstb* gene alone is unlikely to modify APP/Aβ pathogenesis in the mouse model used. Moreover, that the increase in *CSTB* copy number, as occurs in DS, has little impact on the regulation of cathepsin B activity in the brain. Thus, suggesting that this gene does not have a prominent role in modulating AD-DS pathogenesis. However, further studies are required to investigate if 3-copies of *CSTB* affects other aspects of disease suggest as neuronal loss or formation of neurofibrillary tangles. Three copies of other genes on chromosome 21 likely cause the reported differences in endo-lysosomal biology and alterations to proteostasis that occur in DS.

## Material and methods

### Mouse breeding and husbandry

Mice with a duplication of *Cstb* (Cstb^tm2Yah, MGI 5828767, named here dupCstb) were re-derived from embryos and maintained in a colony by mating dupCstb males to C57BL/6J females. J20 mice (Zbtb20^Tg(PDGFB-APPSwInd)20Lms, MGI 3057148, named here tgAPP) were obtained from a colony maintained by mating J20 males to C57BL/6J females. DupCstb females were mated with J20 males to produce the dupCstb x J20 colony. This colony produced mice with four genotypes referred to as: wildtype (WT), dupCstb, tgAPP, and dupCstb*tgAPP.

Genotyping of mice was outsourced to Transnetyx. qPCR was performed to check for any reduction in human *APP* copy number in the tgAPP mice, to exclude from analysis any with a copy number dropped by at least 40% as compared to a J20 positive control DNA, from the Jackson laboratory, Bar Harbor, Maine.

The mice involved in this study were housed in controlled conditions in accordance with Medical Research Council guidance (*Responsibility in the Use of Animals for Medical Research*, 1993), and experiments were approved by the Local Ethical Review panel and conducted under License from the UK Home Office. Cage groups and genotypes were semi-randomised, with a minimum of two mice to a cage; group weaned with members of the same sex. Mouse houses, bedding and wood chips, and continual access to water were available to all mice, with RM1 and RM3 chow (Special Diet Services, UK) provided to breeding and stock mice, respectively. Cages were individually ventilated in a specific-pathogen-free facility. Mice were euthanised by exposure to a rising concentration of $CO_2$ gas, according to the Animals (Scientific Procedures) Act issued in the United Kingdom in 1986.

## Histology

Immediately following euthanasia, the brain was removed, dissected sagittally along the midline and the left hemisphere immerse fixed in 10% buffered formal saline for 48–72 hours (Pioneer Research Chemicals, UK). The fixed tissue was embedded in paraffin wax using an Automated Vacuum Tissue Processor (Leica ASP 300S, Germany) and a series of 4μm sections with the hippocampal formation were cut and mounted onto Superfrost plus glass slides. For amyloid-β immunostaining sections were dewaxed, rehydrated through an alcohol series to water, pre-treated with 80% formic acid for 8mins followed by washing in distilled water for 5mins. The sections were stained for amyloid-β using the Ventana Discovery XT automated stainer, where further pre-treatment (mild CC1–30 minutes of EDTA Boric Acid Buffer, pH 9.0) and blocking (8mins (Superblock, Medite, #88-4101-00), were performed prior to primary antibody incubation (12hrs, biotinylated mouse monoclonal antibody, Sigma-Aldrich SIG-39240 Beta-Amyloid—4G8) at 2μg/ml (antibody diluent, Roche, Switzerland). The staining was completed with the Ventana XT DABMap kit and a haematoxylin counterstain, followed by dehydration and permanent mounting with DPX. All images were acquired using a Leica SCN400F slide scanner analysed using Definiens Tissue Studio and Developer software, with regions of interest manually outlined with reference to a mouse brain atlas [45]. A single operator performed segmentation of all the images, which the software then processed to quantify the area of DAB staining.

## Quantitative reverse transcriptase PCR (qRT-PCR) for *Cstb*

Cortical RNA was extracted as per the miRNeasy Mini Kit protocol (QIAGEN, January 2011), and cDNA was then generated using a QuantiTect Reverse Transcription Kit (QIAGEN), including genomic DNA (gDNA) elimination. TaqMan® quantification using a VIC-*Actb* probe (#4331182) or VIC-*Gapdh* probe (#4331182) and FAM-*Cstb* probe (#451372) was undertaken using a 7500 Fast Real Time PCR System (Applied Biosystems. A 2-fold serial dilution sample of WT mouse cDNA was used as a standard.

## Western blotting

For analysis of protein abundance, cortex was dissected under ice-cold PBS before snap freezing. Samples were then homogenized in RIPA Buffer (150 mM sodium chloride, 50 mM Tris, 1% NP-40, 0.5% sodium deoxycholate, 0.1% sodium dodecyl sulphate) plus complete protease inhibitors (Calbiochem) by mechanical disruption. Total protein content was determined by Bradford assay. Samples from individual animals were run separately and were not pooled.

Cortical homogenates were denatured in NuPAGE® LDS Sample Buffer (Life Technologies, USA) and 2μl 2% β-mercaptoethanol (Sigma-Aldrich) at 100˚C for 5 minutes and separated by SDS-polyacrylamide gel electrophoresis on a NuPAGE® Novex® 4–12% Bis-Tris at

200V for 30 minutes. The proteins in the gel were transferred to a nitrocellulose membrane by Trans-Blot® Turbo™ Transfer System (Bio-Rad Laboratories) at 25V, 2.5A, for 15 minutes. The membranes were blocked with 5% (w/v) milk in PBST (PBS with 0.05% Tween-20 (Sigma-Aldrich)) or Intercept® (PBS) Blocking Buffer (Selected P/N: 927–70001, LI-COR, USA) for 1 hour at room temperature. The membranes were then incubated in primary antibodies overnight at 4˚C, an anti-β-actin mouse monoclonal antibody (Sigma-Aldrich, #A5441, 1:5,000), a rabbit anti-APP antibody (Sigma, USA, #A8717, 1:4,000), an anti- Cystatin B rat monoclonal antibody (Novus Biologicals, USA, #227818, 1:2,000) or a rabbit monoclonal anti-cathepsin B antibody [EPR21033] (Abcam, ab214428, 1:1,000)Followed-by Goat anti-Rat IgG (H+L) Secondary Antibody HRP (Thermo Scientific, USA, # 31470 1:5,000), Goat anti-Rabbit IgG H&L (IRDye® 800CW, Abcam, ab216773, 1:10,000) or Goat anti-Mouse IgG H&L (IRDye® 800CW, Abcam, ab216772, 1:10,000) or Goat anti-Mouse IgG H&L (IRDye® 680RD, Abcam, ab216776, 1:10,000) for 1 hour at room temperature. Once antibody probing was complete, for membranes labelled with secondary antibody HRP, SuperSignal™ West Pico Substrate (Thermo Scientific) was used for chemiluminescent detection of bound proteins, the membranes were exposed to Hyperfilm ECL (GE Healthcare Life Sciences, USA, #10607665) for 30 seconds, and visualised on a ChemiDoc MP Imaging System (Bio-Rad). Membranes labelled with IRDye® 800CW were visualised using Odyssey CLx Infrared Imaging System (LI-COR). The density of protein bands was analysed with Image-J. The calculated density of the band corresponding to CSTB was divided by the density of the β-actin band. To normalise the data to each blot and facilitate comparison of data between blots, the average of two WT relative CSTB values on each blot was taken, and the density values of all lanes on that blot was divided by the WT average. Full uncropped images of the western blots have been deposited at Figshare https://doi.org/10.6084/m9.figshare.14434895.v1.

## Biochemical fractionation and Meso Scale Discovery Assay

Weighed cortex was homogenised in Tris-buffered saline (TBS) (50 mM Tris-HCl pH 8.0) plus complete protease and phosphatase inhibitors (Calbiochem) before centrifugation at 175,000 x g for 30 minutes at 4˚C. The supernatant was removed, snap frozen, and stored at -80˚C as the Tris-soluble fraction. The remaining pellet was re-suspended by homogenising in ice cold 1% Triton-X (Sigma-Aldrich) in TBS (pH 8.0) and centrifuged at 175,000 x g for 30 minutes at 4˚C. The resultant supernatant was removed, snap frozen, and stored at -80˚C as the Triton-soluble fraction. The pellet was re-homogenised in 500μl ice cold 5M Guanidine (Gnd) hydrochloride (Sigma-Aldrich) in TBS (pH 8.0). The final volume was brought to 8x the half-cortex weight with 5M Gnd-TBS, and left rocking overnight at 4˚C to fully re-suspend the sample. This fraction was then snap frozen and stored at -80˚C as the Gnd-soluble fraction. Each biochemical fraction was individually assayed for levels of $A\beta_{38}$, $A\beta_{40}$, and $A\beta_{42}$ using an MSD® MULTI-SPOT Human (6E10) Abeta Triplex Assay (MesoScale Discovery, USA, #K15148E-2) in duplicate. The protocol was carried out as per manufacturer instructions, using reagents provided with the kit; samples were diluted 1:1 (for Tris- and Triton-soluble fractions) or 1:20 (Gnd-soluble fraction) in Diluent 35. Peptide abundance was normalised to wet weight of cortex.

## Cathepsin B activity assay

Cathepsin B activity was assayed in the cortex of 3 or 6-month-old mice using a fluorometric kit (Abcam, #ab65300). Tissue was homogenised in lysis buffer then incubated for 30 minutes on ice, then prior to centrifugation at 15,000 x g for 5 minutes at 4˚C. The supernatant was removed and transferred to a clean tube, on ice, and the protein concentration was determined

by a Bradford assay. For each sample, a volume of supernatant containing 200μg protein used for each reaction with cathepsin B Substrate (RR-amino-4-trifluoromethyl coumarin (AFC)). To control for non-specific cleavage, negative controls treated with inhibitor ALLM (Abcam, ab141446), were run for every sample. Recombinant human cathepsin B (R&D Systems, Cat. No. 953-CY-010) was used as a positive control (0.01 μg for each control reaction). The reaction was incubated at 37°C and the fluorescent output (excitation/emission = 400/505nm) was recorded every 90 seconds for 30 cycles. The linear range of the reaction was determined and the relative cathepsin B activity in the sample calculated by taking the average fluorescent output for each sample minus that in the inhibited. Values are expressed as a percentage of WT average.

## Experimental design and statistical analysis

All experiments and data analysis were undertaken blind to genotype and sex of the mouse, by assignment of a unique 6-digit identifier to all animals and resultant samples. Individual mice were treated as the experimental unit. Study design and group sizes were calculated using effect sizes in (15), using Java Applets for Power and Sample Size, from http://www.stat.uiowa.edu/~rlenth/Power. Graphs were plotted using Prism8 (GraphPad). All statistical analysis was carried out using SPSS Statistics 22 (IBM). Univariate or repeated measures analysis of variance (ANOVA) analyses were used to assess the contribution of tgAPP, dupCstb, and sex of the mouse, to the outcome variable, as well as any interaction between these factors.

## Supporting information

**S1 Fig. Schematic diagram of mouse models trisomic for Hsa21 orthologous genes adapted from Choong et al. (2015).** The regions of Mmu16, 17, and 10 are aligned with their corresponding regions on the long arm of Hsa21, along a megabase pair (Mb) scale. The Tc1 mouse model is represented with breakpoints excluding the Hsa21 genes that are not functionally expressed. The approximate position of the human and mouse APP and CSTB genes are indicated with arrows.
(TIF)

## Acknowledgments

LonDownS Consortium comprises of Andre Strydom (andre.strydom@kcl.ac.uk)[1,2], Elizabeth M.C. Fisher[3], Frances K. Wiseman[4], Dean Nizetic[5,6], John Hardy[3,7], Victor L. J. Tybulewicz[8,9] and Annette Karmiloff-Smith[10]. [1]Department of Forensic and Neurodevelopmental Sciences, Institute of Psychiatry, Psychology and Neuroscience, King's College London, London, UK. [2]Division of Psychiatry, University College London, London, UK. [3]Department of Neuromuscular Diseases, Queen Square Institute of Neurology, University College London, Queen Square, London, UK. [4]The UK Dementia Research Institute, University College London, Queen Square, London, UK. [5]Blizard Institute, Barts and the London School of Medicine, Queen Mary University of London, London, UK. [6]Lee Kong Chian School of Medicine, Nanyang Technological University, Singapore, Singapore. [7]Reta Lila Weston Institute, Institute of Neurology, University College London, London, London, UK. [8]The Francis Crick Institute, London, UK. [9]Department of Immunology and Inflammation, Imperial College, London, UK. [10]Birkbeck University, London, UK.
    We thank Dr. Amanda Heslegrave (UCL-DRI) for assistance with this project.

F.K.W. has undertaken consultancy for Elkington and Fife Patent Lawyers unrelated to the work in the manuscript and is also a PLoS One Academic Editor. This does not alter our adherence to PLoS One policies on sharing data or materials.

## Author Contributions

**Conceptualization:** Elizabeth M. C. Fisher, Frances K. Wiseman.

**Formal analysis:** Yixing Wu, Suzanna Noy.

**Funding acquisition:** Yann Herault, Elizabeth M. C. Fisher, Frances K. Wiseman.

**Investigation:** Yixing Wu, Heather T. Whittaker, Suzanna Noy, Karen Cleverley.

**Resources:** Veronique Brault, Yann Herault.

**Supervision:** Frances K. Wiseman.

**Writing – original draft:** Yixing Wu, Heather T. Whittaker.

**Writing – review & editing:** Heather T. Whittaker, Suzanna Noy, Karen Cleverley, Veronique Brault, Yann Herault, Elizabeth M. C. Fisher, Frances K. Wiseman.

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
