## [Decision Letter · Decision Letter 0]

8 Dec 2020

PONE-D-20-34809

The effects of CSTB duplication on APP/amyloid-β pathology and cathepsin activity in a mouse model

PLOS ONE

Dear Dr. Wiseman,

Thank you for submitting your manuscript to PLOS ONE. After careful consideration by 2 Reviewers and an Academic Editor, all of the critiques of both Reviewers (especially relating to the presentation and interpretation of the Figure 1 as outlined by Reviewer #2 and Reviewer #1) must be addressed in detail in a revision to determine publication status. 

If you are prepared to undertake the work required, I would be pleased to reconsider my decision, but revision of the original submission without directly addressing the critiques of the 2 Reviewers does not guarantee acceptance for publication in PLOS ONE.

If the authors do not feel that the queries can be addressed, please consider submitting to another publication medium. A revised submission will be sent out for re-review. The authors are urged to have the manuscript given a hard copyedit for syntax and grammar.

**Comments to the Author**

1. Is the manuscript technically sound, and do the data support the conclusions?

Reviewer #1: Partly

Reviewer #2: Partly

2. Has the statistical analysis been performed appropriately and rigorously? 

Reviewer #1: I Don't Know

Reviewer #2: Yes

3. Have the authors made all data underlying the findings in their manuscript fully available?

Reviewer #1: Yes

Reviewer #2: No

4. Is the manuscript presented in an intelligible fashion and written in standard English?

Reviewer #1: Yes

Reviewer #2: Yes

5. Review Comments to the Author

Reviewer #1: REVIEW FOR: PLoS One No. PONE-D-20-34809 The effects of CSTB duplication on APP/amyloid-β pathology and cathepsin activity in a mouse model

This is a timely paper on a study that addresses a very important issue in the field of Down syndrome as well as dementia science. It is vital to understand why the DS trisomy of chromosome 21 greatly increases the risk of developing Alzheimer's disease, particularly since gene(s) on chromosome 21 other than APP were shown to accelerate amyloid pathology when triplicated. To determine if three copies of CSTB, the endogenous inhibitor cystatin B of lysosomal cysteine cathepsins that is located on chromosome 21, influences Alzheimer-type pathogenesis a tgAPP mouse model of amyloid-β deposition was crossed with a mouse with heterozygous duplication of the Cstb locus. The Cstb duplication increased transcriptional and translational levels of CSTB in the brain, while having no detected effect on plaque deposition or Aβ levels. With the following additional analyses of immunoblots and enzyme assays, along with minor improvements to the text and references, this will be an important study to help understand the DS-AD connection.

1. A clear statement of the hypothesis being tested needs to be included in the abstract and/or introduction, and reflected by the title. This point is made since there exists several (~11) cysteine cathepsin enzymes, and the abstract states that duplication of the gene Cstb in a transgenic APP mouse model resulted in an increase in transcriptional and translational levels of Cstb but had no effect on Aβ or plaques… and the increased CSTB did not alter the activity of cathepsin B enzyme. Thus, it should be clearly stated the selectivity of the hypothesis, i.e. we determined if three copies of the mouse gene Cstb is sufficient to modulate Aβ accumulation and cathepsin B activity in a transgenic APP mouse model. Also, as Cstb duplication is noted as increasing transcription levels of Cstb in tg mice, please address why no statistical significance is noted in Figure 1A for tgAPP vs. dupCstb*tgAPP (also no such notation between those two groups in Fig. 1B).

2. With cathepsin B being the focus of the tested hypothesis, and that establishing a negative result requires more extensive analysis than to establish a positive change, additional background information and data are warranted. For instance, Mueller-Steiner et al. (ref. 22) indicate that message, protein, and activity levels of cathepsin B are increased by Aβ-related proteinopathy, but the putative compensatory response failed to occur in older tg mice which may be related to the aging risk factor for AD. Such should be mentioned in the context of whether CSTB plays a role in cathepsin B’s compensatory response and related regulation of the enzyme’s maturation process. And, since the ages of tg mice used in the Wu et al. study cannot make such comparison as in the Mueller-Steiner study, measures of cathepsin B maturation (active CatB or actCatB:proCatB ratio) can be assessed in immunoblot samples to enhance the study, and/or the specific assessment of lysosomal cathepsin B activity in isolated lysosomes via Percoll step as previously shown (Butler et al. 2011 PLoS ONE 6:e20501). Such analyses would provide improved coverage of whether Cstb regulation influences key features of cathepsin B maturation, trafficking, and localized activity that have been shown linked to AD-type pathology and may be critical for the DS-AD connection.

3. The anti-APP immunoblot in Figure 1D should be shown in full for purpose of assessing CTFs (or lack thereof if a positive control that the antibody recognizes them). Relevance on this matter from “Lysosomal Dysfunction in Down Syndrome is Mediated by APP-βCTF” by Nixon’s lab (Jiang et al. 2019 J Neurosci 39:5255) and others indicating APP-CTFs-derived endosomal dysfunction is early hallmark of both AD and DS (Pérez-González et al. 2019 Neurobiol Aging 84:26). Also, enhancing cathepsin B reduced APP-CTFs in AD mice (Hwang et al. 2019 Internatl J Mol Sci 20(18):4432). Thus up-regulated CSTB would be expected to increase APP-CTFs.

4. Discussion could include brief mention of the crosstalk between the lysosomal cathepsin network and the proteasome pathway as another potential component of the complex DS-AD relationship, as the two protein clearance systems may both contribute to the proteostatic stress in Down syndrome. See Aivazidis et al. 2017 Plos ONE 12(4):e0176307 and Farizatto et al. 2017 PLoS ONE 12(8): e0182895.

Lastly, minor but important reference coverage and corrections are warranted in the Introduction. Ref. 24 is noted as showing that knocking out Cstb lowered cathepsin B activity (typo?). This is a problem with logic for the Wu study and misrepresents Yang et al. that clearly says Cystatin B deletion in TgCRND8 elevating cathepsin activities, results of enzymatic activity for cathepsin B.

Several cathepsin B studies noted above would also add for a more appropriate coverage of the important topic. As for Ref. 21 regarding “cathepsin B inhibitors CA074Me or E64d lead to reduced Aβ levels”, please note that E64d is well-known for blocking the protease calpain more potently than cathepsin B, and that E64d is reported to produce complete neuroprotection in cathepsin B knockout mice (Hook et al. 2014). As for CA074me, it has been shown to inhibit cathepsin B without having inhibitory action on human b-secretase (Butler et al. 2011).

Reviewer #2: Cystatin B (CSTB) is an endogenous inhibitor of cathepsin proteases, the current study aimed to evaluate the effects of duplication of CSTB on amyloid pathology using dupCstBXtgAPP mice. The authors found CSTB duplication resulted in an increase in the expression of CSTB in the cortex from 3-month old mice, but had no effects on the enzymatic activity on cathepsin B, the endogenous substrate of CSTB. Furthermore, CSTB duplication had no effects on neither the soluble nor insoluble Aβ in 6-month old tgAPP mice. The study is relatively simple and clear; however, revisions are indicated.

Major comments:

1) The effects of CSTB duplication on APP processing were examined using 6-month old mice. It will be reasonable to evaluate the expression of CSTB and enzymatic activity of cathepsin B using the same sample, but not use the sample from 3-month old mice.

2) According to figure1 D and E, the expression of APP was obvious changed in dupCstB*tgAPP mice compared to tgAPP mice in figure. 1D, but no difference was shown in figure.1E.

3) If the inhibition of cathepsins has been saturated by endogenous CSTB in 6-month old tgAPP mice, duplication of CSTB was destined to have no effects on relative pathologies. Consideration and discussion on this point is lacking.

4) A positive control using human recombinant cathepsin B in the CatB activity assay is absolutely needed.

Again, the authors should make an effort for exploring the effects of Cystatin B on AD pathologies, not limited to Amyloid beta.

6. PLOS authors have the option to publish the peer review history of their article (what does this mean?). If published, this will include your full peer review and any attached files.

**Do you want your identity to be public for this peer review?** For information about this choice, including consent withdrawal, please see our Privacy Policy.

Reviewer #1: No

Reviewer #2: No

We look forward to receiving your revised manuscript.

Kind regards,

Stephen D. Ginsberg, Ph.D.

Section Editor

PLOS ONE

2.Thank you for stating the following in the Competing Interests section:

[F.K.W. has undertaken consultancy for Elkington and Fife Patent Lawyers unrelated to the work in the manuscript and is also a PLoS One Academic Editor.].

3.PLOS ONE now requires that authors provide the original uncropped and unadjusted images underlying all blot or gel results reported in a submission’s figures or Supporting Information files. This policy and the journal’s other requirements for blot/gel reporting and figure preparation are described in detail at https://journals.plos.org/plosone/s/figures#loc-blot-and-gel-reporting-requirements and https://journals.plos.org/plosone/s/figures#loc-preparing-figures-from-image-files. When you submit your revised manuscript, please ensure that your figures adhere fully to these guidelines and provide the original underlying images for all blot or gel data reported in your submission. See the following link for instructions on providing the original image data: https://journals.plos.org/plosone/s/figures#loc-original-images-for-blots-and-gels.

4. One of the noted authors is a group or consortium [LonDownS Consortium]. In addition to naming the author group, please list the individual authors and affiliations within this group in the acknowledgments section of your manuscript. Please also indicate clearly a lead author for this group along with a contact email address.

---

## [Author Response · Author response to Decision Letter 0]

27 Apr 2021

PONE-D-20-34809

The effects of CSTB duplication on APP/amyloid-β pathology and cathepsin activity in a mouse model

We thank the editor and reviewers for their constructive and helpful feedback and have revised our manuscript to take account of all comments including those relating to the presentation of data in Figure 1, please find a detailed response below.

We have also now deposited the original western blot images, for the data represented in Fig 1 C, E and F and Figure 4D at Figshare (DOI:10.6084/m9.figshare.14434895), please see https://figshare.com/s/05d31651d620ce5ceaea. We will update this file with details of the paper if it is accepted for publication at PLoS One and publish. 

We have updated the competing interests statement to now read. “F.K.W. has undertaken consultancy for Elkington and Fife Patent Lawyers unrelated to the work in the manuscript and is also a PLoS One Academic Editor. This does not alter our adherence to PLoS ONE policies on sharing data and materials”. 

We have also updated the acknowledgments to provide full-details of the LonDownS consortium.

Yours sincerely

Frances Wiseman

Detailed response to reviewers

Reviewer one

1. A clear statement of the hypothesis being tested needs to be included in the abstract and/or introduction, and reflected by the title. 

We have revised our title to “The effects of CSTB duplication on APP/amyloid-β pathology and cathepsin B activity in a mouse model and updated our abstract.

2. With cathepsin B being the focus of the tested hypothesis, and that establishing a negative result requires more extensive analysis than to establish a positive change, additional background information and data are warranted. For instance, Mueller-Steiner et al. (ref. 22) indicate that message, protein, and activity levels of cathepsin B are increased by Aβ-related proteinopathy, but the putative compensatory response failed to occur in older tg mice which may be related to the aging risk factor for AD. Such should be mentioned in the context of whether CSTB plays a role in cathepsin B’s compensatory response and related regulation of the enzyme’s maturation process. And, since the ages of tg mice used in the Wu et al. study cannot make such comparison as in the Mueller-Steiner study, measures of cathepsin B maturation (active CatB or actCatB:proCatB ratio) can be assessed in immunoblot samples to enhance the study, and/or the specific assessment of lysosomal cathepsin B activity in isolated lysosomes via Percoll step as previously shown (Butler et al. 2011 PLoS ONE 6:e20501). Such analyses would provide improved coverage of whether Cstb regulation influences key features of cathepsin B maturation, trafficking, and localized activity that have been shown linked to AD-type pathology and may be critical for the DS-AD connection.

We now present additional activeCatB/proCatB data in wildtype and dupCstb mice Fig 4D and have updated the discussion please see manuscript lines 176-179.

3. The anti-APP immunoblot in Figure 1D should be shown in full for purpose of assessing CTFs (or lack thereof if a positive control that the antibody recognizes them). Relevance on this matter from “Lysosomal Dysfunction in Down Syndrome is Mediated by APP-βCTF” by Nixon’s lab (Jiang et al. 2019 J Neurosci 39:5255) and others indicating APP-CTFs-derived endosomal dysfunction is early hallmark of both AD and DS (Pérez-González et al. 2019 Neurobiol Aging 84:26). Also, enhancing cathepsin B reduced APP-CTFs in AD mice (Hwang et al. 2019 Internatl J Mol Sci 20(18):4432). Thus up-regulated CSTB would be expected to increase APP-CTFs.

We reran our western blots to also study APP-C-terminal fragments in addition to full-length APP protein, please see Fig. 1 (updated) D and F. We have also updated the result section to describe these data manuscript. Lines no. 122-126. We note that an additional copy of Cstb in our mouse model is not sufficient to alter APP-CTF abundance.

4. Discussion could include brief mention of the crosstalk between the lysosomal cathepsin network and the proteasome pathway as another potential component of the complex DS-AD relationship, as the two protein clearance systems may both contribute to the proteostatic stress in Down syndrome. See Aivazidis et al. 2017 Plos ONE 12(4):e0176307 and Farizatto et al. 2017 PLoS ONE 12(8): e0182895.

We have updated the discussion to further highlight the complexity of AD-DS mechanisms, please see manuscript lines 232-236.

Lastly, minor but important reference coverage and corrections are warranted in the Introduction. Ref. 24 is noted as showing that knocking out Cstb lowered cathepsin B activity (typo?). This is a problem with logic for the Wu study and misrepresents Yang et al. that clearly says Cystatin B deletion in TgCRND8 elevating cathepsin activities, results of enzymatic activity for cathepsin B.

We apologies for the error in our original submission regarding the description of the data in Yang et al “Knocking out Cstb by crossing the Cstbtm1Rm (23) and an APP transgenic mouse model (TgCRND8) lowered cathepsin B activity------ the word “lowered” was a typo and has been corrected in the updated manuscript. We also have expanded our discussion regarding the differing phenotypic effects of increased and decreased gene dose; please see manuscript lines 182-193.

Several cathepsin B studies noted above would also add for a more appropriate coverage of the important topic. As for Ref. 21 regarding “cathepsin B inhibitors CA074Me or E64d lead to reduced Aβ levels”, please note that E64d is well-known for blocking the protease calpain more potently than cathepsin B, and that E64d is reported to produce complete neuroprotection in cathepsin B knockout mice (Hook et al. 2014). As for CA074me, it has been shown to inhibit cathepsin B without having inhibitory action on human b-secretase (Butler et al. 2011).

We have updated manuscript the include this literature, please see lines 77-91.

Reviewer two

1) The effects of CSTB duplication on APP processing were examined using 6-month old mice. It will be reasonable to evaluate the expression of CSTB and enzymatic activity of cathepsin B using the same sample, but not use the sample from 3-month old mice.

To address this comment, we have repeated the study of cathepsin B activity at 6-months of age (Fig. 4B), duplication of Cstb did not modulate enzyme activity at this age in the cortex of our mouse model. We have added a description to these data to the manuscript; lines 169-175.

2) According to figure1 D and E, the expression of APP was obvious changed in dupCstB*tgAPP mice compared to tgAPP mice in figure. 1D, but no difference was shown in figure.1E.

Figure 1D was a representative western blot (n=1); whereas our conclusion is based on statistical analysis results (n=5 for each genotype). We have therefore updated figure 1D using a western blot that can better represent our results. We note all western blots for this study have been uploaded to FigShare (DOI:10.6084/m9.figshare.14434895).

3) If the inhibition of cathepsins has been saturated by endogenous CSTB in 6-month old tgAPP mice, duplication of CSTB was destined to have no effects on relative pathologies. Consideration and discussion on this point is lacking.

We have added a discussion of this comment, please see the updated manuscript lines 220-223.

4) A positive control using human recombinant cathepsin B in the CatB activity assay is absolutely needed.

We included this positive control (Fig 4B) and the updated figure legend in the manuscript.

Again, the authors should make an effort for exploring the effects of Cystatin B on AD pathologies, not limited to Amyloid beta.

We agree that the effect of Cstb duplication on other aspects of AD pathology would be extremely interesting to investigate but this is out with the scope of the data represented in this report. We have added this comment to the discussion, lines 235-236.

---

## [Decision Letter · Decision Letter 1]

17 May 2021

The effects of CSTB duplication on APP/amyloid-β pathology and cathepsin B activity in a mouse model

PONE-D-20-34809R1

Dear Dr. Wiseman,

We’re pleased to inform you that your manuscript has been judged scientifically suitable for publication and will be formally accepted for publication once it meets all outstanding technical requirements.

Kind regards,

Stephen D. Ginsberg, Ph.D.

Section Editor

PLOS ONE

**Comments to the Author**

1. If the authors have adequately addressed your comments raised in a previous round of review and you feel that this manuscript is now acceptable for publication, you may indicate that here to bypass the “Comments to the Author” section, enter your conflict of interest statement in the “Confidential to Editor” section, and submit your "Accept" recommendation.

Reviewer #1: All comments have been addressed

2. Is the manuscript technically sound, and do the data support the conclusions?

Reviewer #1: Yes

3. Has the statistical analysis been performed appropriately and rigorously? 

Reviewer #1: Yes

4. Have the authors made all data underlying the findings in their manuscript fully available?

Reviewer #1: Yes

5. Is the manuscript presented in an intelligible fashion and written in standard English?

Reviewer #1: Yes

6. Review Comments to the Author

Reviewer #1: Authors have addressed all my concerns regarding CatB-related importance for involvement in neuropathogenic insults.

7. PLOS authors have the option to publish the peer review history of their article (what does this mean?). If published, this will include your full peer review and any attached files.

Reviewer #1: No

---

## [Editor Report · Acceptance letter]

13 Jul 2021

PONE-D-20-34809R1 

The effects of *Cstb* duplication on APP/amyloid-β pathology and cathepsin B activity in a mouse model 

Dear Dr. Wiseman:

I'm pleased to inform you that your manuscript has been deemed suitable for publication in PLOS ONE. Congratulations! Your manuscript is now with our production department. 

Kind regards, 

on behalf of

Dr. Stephen D. Ginsberg 

Section Editor

PLOS ONE